# Corrosion Resistance of Zinc and Zinc-Aluminum-Magnesium Coatings in Atmosphere on the Territory of Russia

**DOI:** 10.3390/ma16155214

**Published:** 2023-07-25

**Authors:** Yulia M. Panchenko, Andrey I. Marshakov, Timofey N. Igonin, Tatyana A. Nenasheva, Ludmila A. Nikolaeva, Artem A. Ivanenko

**Affiliations:** 1A.N. Frumkin Institute of Physical Chemistry and Electrochemistry, Russian Academy of Sciences, 31 Leninskii Prospect, Moscow 119071, Russia; panchenkoyum@mail.ru (Y.M.P.); igonintd@gmail.com (T.N.I.); nenasheva@ipc.rssi.ru (T.A.N.); igonin@ipc.rssi.ru (L.A.N.); 2Novolipetsk Steel, Metallurgov Sq., 2, Lipetsk 398040, Russia; ivanenko_aa@nlmk.com

**Keywords:** atmospheric corrosion, hot dip galvanized steel, zinc-aluminum-magnesium coatings, atmosphere corrosivity categories, modeling, service life

## Abstract

Zinc-coated carbon steel is commonly used in the construction of buildings, infrastructure objects such as roads and bridges, automotive production, etc. Coatings based on zinc-aluminum-magnesium alloys that may have better corrosion resistance than zinc have been developed. The coatings made of the new alloys have been available on the market for a shorter period of time than conventional zinc coatings. This paper presents data on the corrosion resistance of zinc and zinc-aluminum-magnesium coatings on carbon steel obtained by tests in four locations in Russia with marine and non-marine atmospheres. Four one-year exposures at the beginning of each season and two-year tests were performed. It is shown that the corrosion resistance of the coatings depends significantly on the beginning of the exposure. The categories of atmosphere corrosivity in relation to the coatings were determined at each location. Based on the dose–response function (DRF) for zinc developed for the territory of Russia, DRFs for the coatings were obtained. A match between the categories of atmosphere corrosivity determined by the first-year corrosion losses and estimated from the values of corrosion losses calculated using the DRF is shown. Based on the data of two-year tests, the variation in the corrosion rate over time is obtained. The corrosion rates of the coatings in the territory of Russia are compared to the corrosion rates of coatings observed in various locations around the world. An approximate estimate of the service life of the coatings at the test sites is given.

## 1. Introduction

Steels of various types are the most widely used construction materials due to their mechanical strength and low cost. They are usually employed for objects that are exposed to the action of the ambient atmosphere. The most efficient method of protection of structures against corrosion and for the extension of their service life involves the use of anticorrosion coatings, including organic and metal coatings. Among metal coatings, various types of zinc coatings are the most widely used. The corrosion protection mechanism of such coatings involves the barrier effect of the metal layer itself, galvanic protection as a sacrificial anode, and the barrier effect of corrosion products that inhibit the corrosion of the metal layer [1,2,3,4,5]. At the early stages of the use of metal structures, the barrier-related effect and galvanic protection are the main mechanisms that prevent the corrosion of steel. The galvanic effect of a coating is most efficient at locations where the metal layer has been significantly damaged (for example, scratched) or does not cover the entire surface of a structure (for example, the cut edges of steel sheets) [5]. Galvanized steel is widely used in the construction of buildings and in infrastructure facilities such as roads, bridges, etc. Zinc coatings can extend the service life of steel structures and reduce the cost of anti-corrosion measures during their use.

In recent years, a wide range of metal coatings have been developed, such as zinc-aluminum and zinc-aluminum-magnesium alloy coatings. These coatings should feature better corrosion resistance while reducing the cost of products due to a reduction in the coating mass per unit of surface area. These alloys are recommended by manufacturers primarily for applications in which thick zinc coatings have traditionally been used to withstand the effects of corrosion in harsh environments.

For zinc-based coatings, a wealth of disparate information is available regarding their corrosion resistance in various outdoor climates around the world. Data on the corrosion resistance of coatings with zinc alloys, obtained in 36 locations worldwide and presented in [6,7,8,9,10,11,12,13,14,15,16,17,18], are summarized and systematized in [19]. In this work, field test data are systematized in the following ways: by corrosion damage to alloy coatings in accordance with the categories developed for zinc and by the change in corrosion damage to coatings and their corrosion rates over time for each category. Based on the analysis of the generalized results of full-scale tests for categories C2, C3, and C4, the ranges of corrosion damage rates for individual groups of coatings with alloys for 2, 4, and 6 years of exposure are presented.

ZnAlMg coatings have not been tested before in the territory of Russia. The purpose of the study is to assess the corrosion resistance of ZnAlMg coatings in four representative locations in Russia; to determine the categories of atmosphere corrosivity in relation to them; to develop the dose–response function (DRF) for the calculation of the first-year corrosion losses of the coatings; and to roughly estimate the service life of the coatings.

## 2. Materials and Methods

### 2.1. Materials

Samples of steel 08ps with a hot-dipped ZnAlMg coating and a hot-dipped galvanized steel coating (HDG coating) were tested. In addition, samples of steel 08ps (%: C 0.05–0.11; Si 0.05–0.17; Mn 0.35–0.65; P < 0.03; S < 0.035; Cr < 0.1; Ni < 0.3. Cu < 0.3) without a coating and sheet zinc (no less than 99.95% Zn) were tested. Steel 08ps was similar to steel EN 10130 DC03. The elemental composition of coatings and their nominal masses are presented in Table 1.

The density of the ZnAlMg coating is 6.45 ± 0.05 g cm^−3^. The nominal thicknesses of HDG and ZnAlMg coatings are 19 and 9 µm, respectively. The HDG coating is uniform in thickness and composition, whereas the ZnAlMg coating has a multiphase structure: primary phases of zinc were observed whose grains are separated by eutectic phases; namely, binary eutectic Zn-MgZn_2_ and ternary eutectic mixture Zn-MgZn_2_-Al [20] are found, which agrees with the previously known data on the phase composition of ZnAlMg coatings [21,22].

### 2.2. Testing Conditions

Materials were tested at four corrosion stations (CS) whose coordinates are presented in Table 2 along with a classification of atmospheres.

At each CS, 4 one-year tests were carried out at open areas (OA) and under a shelter (S) with the start of exposure on: 1 June (batch 1), 1 September (batch 2), 1 December (batch 3), and 1 March (batch 4). Concurrently with the installation of the first batch of samples on 1 June, samples were installed for 2 years of exposure. Three samples of uncoated steel and zinc and five samples with ZnAlMg and HDG coatings were installed in each batch. Samples were set at an angle of 45° with respect to the horizon at all the exposure sites. At MCS and ZCS, samples were oriented with their upper side to the south, and at NCS and FECS, towards the nearest seashore.

The samples were prepared and installed in accordance with procedures reported elsewhere [23]. To remove organic contaminants, the surface of all the samples to be tested was degreased with acetone. Before exposure, zinc sheet samples were etched for 10 min at 25 °C in a solution of glycine (aminoacetic acid, NH_2_CH_2_COOH) (250 g·L^−1^). After etching, the samples were washed with distilled water and dried with filter paper.

During the test period, meteorological parameters—temperature (*T*, °C), relative humidity of air (*RH*, %), and total annual rainfall (Rain, mm)—were continuously recorded. Atmospheric pollution with sulfur dioxide (SO_2_, mg·m^−2^·day^−1^) was assessed using sulfate plates, and to assess the rate of chloride deposition on metal samples (Cl^−^, mg·m^−2^·day^−1^), a “wet candle” was used [24]. Both sides of samples with coatings were regularly inspected to identify the appearance of spots of steel corrosion products.

Corrosion products were removed from the samples by a chemical method in accordance with the reported technique [25]. Samples of steel 08ps were etched for 10 min at 25 °C in a solution containing 500 mL (HCl, *ρ* = 1.19 g·cm^−3^) + urotropine (C_6_H_12_N_4_) (3.5 g·L^−1^). Samples made of zinc sheets and coated with ZnAlMg and HDG were etched for 10 min at 25 °C in a glycine solution (250 g·L^−1^). After removing the corrosion products, the samples were washed with distilled water and dried with filter paper. All the solutions were prepared from reagents of “chemically pure” grade and distillate.

Samples were weighed before and after exposure using a ViBRA AF-R220CE analytical balance with an accuracy of 0.0001 g.

### 2.3. Dose–Response Functions

To predict the corrosion losses of materials in the first year of exposure (*K*_1_), dose–response functions (DFRs) were used. A general view of DFRs is given in [26]. For the territory of Russia, DFRs have now been developed [27,28,29], the use of which for typical metals (carbon steel, zinc, copper, aluminum) gives the most reliable predictions of *K*_1_ for open areas [27].

## 3. Results

### 3.1. Meteorological and Chemical Parameters of the Atmospheres

The annual average meteorological and chemical parameters of the atmospheres for various periods are displayed in Table 3.

The ranges of atmosphere corrosivity parameters at the testing sites in Russia were: 2.1–8.2 °C; 66–81%; 220–884 mm/year; [SO_2_] 2.46–8.2 mg·m^−2^·day^−1^; and, at the coastal CS, 14.4–187 [Cl^−^], mg·m^−2^·day^−1^. The broad range of environmental parameters covers the climatic and air chemical conditions in the territory of Russia with sufficient completeness.

The data presented above show that the annual average parameters of atmosphere corrosivity for all batches at each CS are approximately the same, the only exception being FECS, where [Cl^−^] ranges from 14.4 to 36.6 mg·m^−2^·day^−1^ (Table 3). At the three other CS, approximately the same values of corrosion losses in one-year periods could be expected for the four batches.

### 3.2. Determination of the Atmospheric Corrosivity at a CS with Respect to Materials at Open Areas and under a Shelter

The atmosphere corrosivity is determined from the experimental corrosion losses of metals during the first year of exposure [26]. For typical metals, corrosivity categories of the atmosphere from C1 to CX for an open atmosphere are presented. For ZnAlMg and HDG coatings, we used the categories defined for zinc as a first approximation, and for the atmosphere under a shelter, we used the categories for open atmosphere. Taking into account that the atmosphere corrosivity in most of Russia’s territory is in category C2, additional subcategories of C2-1, C2-2, and C2-3 were introduced [30].

To determine the atmosphere corrosivity category at a specific place, several one-year tests are required, taking into account the spread of the average annual values of atmospheric parameters. To reduce the time of testing, we carried out four one-year tests with a displacement in the start of exposure by 3 months, i.e., at each CS in the open atmosphere and under a shelter for all materials, so four values of one-year mass losses (*K*_1_) were obtained. The values of *K*_1_ and the atmosphere corrosivity category for each period of exposure and their average values for the open atmosphere and under a shelter are presented in Table 4 and Table 5, respectively. The data obtained are visualized in Figure 1 (open atmosphere) and Figure 2 (under a shelter). Table 4 and Table 5 also give the values of ±Δ*K*_1_, which were defined as:−∆*K*_1_ = *K*_1,small_ − *K*_1_,    +∆*K*_1_ = *K*_1,large_ − *K*_1_,
where *K*_1,small_, *K*_1,large_, and *K*_1_ are the smallest, largest, and average values of the corrosion losses of the samples tested simultaneously in each batch. The arrows in Figure 1 and Figure 2 show the deviation of *K*_1_ from the average value. As can be seen, the differences in the corrosion losses of simultaneously tested samples in the same batch are less than the differences in the average values of *K*_1_ in four batches (Table 4 and Table 5, Figure 1 and Figure 2).

The results obtained indicate a large corrosion loss of materials at the coastal CS in comparison with those at the non-marine CS. For example, in the open area (OA), the *K*_1_ values of steel differ 15-fold, zinc—5.5-fold, HDG coatings—3.9-fold, and ZnAlMg coatings—7.4-fold, which is due to the difference in climatic and aerochemical parameters at different CS. When the samples are exposed under a shelter (S), the *K*_1_ values differ to a greater extent: by a factor of 24 (steel), 9.8 (zinc), 50 (HDG coatings), and 12.2 (ZnAlMg coatings), which can be explained by the toughening of conditions at the coastal CS due to the accumulation of sea salts on the surface of the samples, which are not washed off by rains.

At each CS, the values of *K*_1_ for the four batches of samples under OA and S conditions have different values corresponding to different atmosphere corrosivity categories or different subcategories of category C2. Such a difference in the *K*_1_ values at approximately equal average annual atmospheric parameters during testing of different batches of samples can be explained by the effect of the starting conditions, i.e., differences in the atmosphere corrosivity during the first months of exposure. To clarify the reasons for the effect of the starting exposure conditions, additional studies are needed that would take the environmental parameters during the first months of testing into account, which is beyond the scope of this work.

The corrosion resistance of coatings can be compared using the average *K*_1_ values for the four batches of samples, although significantly diverging *K*_1_ data for individual batches may affect the results of the comparison. In open area, the ZnAlMg coating is more corrosion-resistant than the HDG coating (with the exception of the data obtained at the NCS), but under a shelter, the HDG coating exhibited smaller corrosion losses in 1 year (with the exception of the FECS).

### 3.3. Accessing the Atmosphere Corrosivity Categories

For practical purposes, the correct assessment of the atmosphere corrosivity category [31] is of greater importance than the accuracy of corrosion rate prediction, since for each category, designers provide for the thickness of the metal of products, methods, and means of protection and preservation, estimate their durability, etc. Without corrosion testing, *K*_1_ can be calculated using dose–response functions (DRFs). Below, we present the DRFs only for *T* ≤ 10 °C, because the temperatures at all the CS do not exceed this threshold [27]:

For carbon steel, Equation (1):*K*_1_ = 7.7·{[SO_2_]^0.47^ + 0.68·[Cl]^0.58^}·exp(0.024·*RH* + 0.095·(*T*-10) + 0.00035·*Prec*)(1)

For zinc, Equation (2):*K*_1_ = 0.45·{[SO_2_]^0.36^ + 0.64·[Cl]^0.51^}·exp(0.023·*RH* + 0.025·(*T*-10) + 0.00035·*Prec*)(2)

It is necessary to determine the applicability of Equations (1) and (2) to the materials under study and, if possible, to make adjustments to the DRF to apply them to coatings exposed outdoors and under a shelter. Taking into account that the range of *K*_1_ values for various one-year sample batches is rather wide, it is first of all necessary to check whether the DRF can be used to obtain a reliable estimate of the categories of atmosphere corrosivity. To do so, it is necessary to estimate the categories based on the predicted *K*_1_ values (*K*_1_^pr^) and compare them with the categories determined from the experimental *K*_1_ values (*K*_1_^ex^). For comparison, all four values of *K*_1_^ex^ for each material were used.

According to the parameters of atmosphere corrosivity for each one-year period of testing, the predicted values of mass losses were calculated by the DRF, i.e., for each material, four values of *K*_1_^pr^ were obtained at each CS. Equation (1) was used to calculate *K*_1_^pr^ for steel 08ps, and Equation (2) was used to calculate the corrosion losses of zinc and coatings.

The results of comparing *K*_1_^pr^ with *K*_1_^ex^ obtained in an open area and under a shelter are shown in Figure 3 and Figure 4, respectively. The location of the points (*K*_1_^ex^; *K*_1_^pr^) on the coordinate field of these plots makes it possible to estimate the reliability of K_1_^pr^: if they coincide with *K*_1_^ex^, the points lie on a solid line *K*_1_^ex^ = *K*_1_^pr^. The dashed lines in the plots show the uncertainty interval of the forecast values according to [26], which for steel and zinc is within a range of relative errors (δ) from −33% to +50%. These figures show the atmosphere corrosivity categories according to [26]. If the categories determined from *K*_1_^ex^ and estimated by *K*_1_^pr^ coincide, the points (*K*_1_^ex^; *K*_1_^pr^) are located inside the squares, limiting the ranges of *K*_1_ values of the corresponding category.

Based on the results obtained (Figure 3 and Figure 4), the following should be noted:−Steel 08ps. Most of the *K*_1_^pr^ values correspond to the atmosphere corrosivity categories determined from *K*_1_^ex^.−Zinc. The values of *K*_1_^pr^ calculated for tests in an open area and under a shelter at the NCS are overestimated, according to which the atmosphere category is C4 instead of the category C3 determined experimentally. The *K*_1_^pr^ values calculated for the OA conditions at the ZCS are underestimated, as a result of which, according to the predictive estimate, the C2 category is obtained instead of C3.−Coatings. The *K*_1_^pr^ values calculated for test conditions in an open area and under a shelter in the marine atmosphere are greatly overestimated, as a result of which the atmosphere category C3 was obtained at FECS instead of C2, and at NCS, category C4 was found instead of C3. In the non-marine atmospheres (MCS and ZCS), the atmospheric category is correctly assessed as C2.

To obtain reliable *K*_1_^pr^ values, the DRF should be adjusted. Given that the SO_2_ concentration in air at all the CS is close to the background value [26] and does not exceed 4 μg m^−3^, the change in the exponent for [SO_2_] has little effect on the value of *K*_1_. It is most likely that the first factor in Equations (1) and (2), denoted as *A,* and the exponent of the parameter [Cl], denoted below as *β,* have to be adjusted.

The DRF for steel 0.8ps was changed insignificantly as follows:*K*_1_ = 7.7·{[SO_2_]^0.47^ + 0.68·[Cl]^0.56^}·exp(0.024·*RH* + 0.095·(*T*-10) + 0.00035·*Prec*)(3)

The DRF for zinc and the coatings is presented in a general form:*K*_1_ = *A*·{[SO_2_]^0.36^ + 0.64·[Cl]^β^}·exp(0.023·*RH* + 0.025·(*T*-10) + 0.00035·*Prec*)(4)

The values of coefficients *A* and *β* for the latter equation are shown in Table 6 for open area and under a shelter.

The atmosphere corrosivity categories estimated from *K*_1_^pr^ calculated using Equations (3) and (4) and determined from *K*_1_^ex^ are compared in Figure 5 and Figure 6 for open area and a shelter, respectively. It can be seen that the use of adjusted values of *A* and *β* coefficients in the DRF provides a more accurate prediction for *K*_1_ and a more accurate estimation of atmosphere corrosivity. It should be noted that Figure 4 and Figure 5 compare the predicted *K*_1_ with *K*_1_^ex^ for all four batches of samples. It is natural that, given relatively identical annual average parameters of atmosphere corrosivity, it is difficult to provide reliable *K*_1_ predictions for all batches, since the effect of the initial exposure conditions yields “outlying” values of *K*_1_^ex^, as it is clearly seen in these figures.

If individual *K*_1_^ex^ outliers are not taken into account, the relative errors (δ) of the calculated *K*_1_^pr^ values are within a permissible range according to [26] (from −33% to +50%), see Table 7. The same table displays the error in *K*_1_^pr^ prediction calculated from the average atmosphere corrosivity parameters for four year-long exposures with respect to the average values of *K*_1_^ex^ for all batches. In this case, the +δ% and −δ% values are usually smaller.

In the models, the values of the coefficients are selected to provide the prediction accuracy, and their values cannot always be explained logically. However, in this case, we make an attempt to explain the *A* and *β* values obtained.

Open area. The correction of *A* and *β* values for zinc may be explained by different amounts of microimpurities in zinc. For example, in full-scale tests conducted under the ISO CORRAG program [32], zinc samples with a purity of at least 99.85% were used, whereas in our tests, zinc with a purity of at least 99.95% was employed. The HDG coating features virtually the same high corrosion resistance at all the CS, the *A* and *β* coefficients being significantly smaller than those for zinc. The ZnAlMg coating compared to the HDG coating is more resistant in non-marine atmospheres but less resistant at NCS, where the chloride deposition rate is high. Therefore, the *A* coefficient is smaller and *β* is larger than for the HDG coating.

Shelter. The corrosion resistance of steel 08ps for the first year under a shelter is virtually the same as in an open area, the values of *A* and *β* being the same as in the open area. The corrosion resistance of zinc and especially of the coatings in a non-marine atmosphere is lower than in open area, which is reflected in a lower value of the *A* coefficient. In marine atmosphere, the *β* coefficients for zinc, HDG, and ZnAlMg120 coatings are higher than that for open area. This is probably due to the fact that during the tests under a shelter, chlorides that deposit on the surfaces of materials are not washed off by rain. Due to the absence of a “wash-off effect” [28], the *β* values are higher.

### 3.4. Corrosion Rate of Coatings for 2 Years of Exposure

The corrosion rates of the materials (σ_2_) were determined from the corrosion losses for 2 years of exposure of samples (*K*_2_): σ_2_ = *K*_2_/2. The values of σ_2_ for the materials in open area and under a shelter are presented in Table 8.

After 2 years of exposure in open area at all the CS, the corrosion rate of ZnAlMg coatings was smaller than that of HDG coatings, whereas under a shelter, it was only observed at NCS and FECS. Comparison of the values of *σ*_2_ (Table 8) and *K*_1_ in the first batch (Table 4 and Table 5) shows that the corrosion rate of all materials decreases after 2 years of exposure, with the only exception being that the corrosion rate of HDG coatings under a shelter at the ZCS has not practically changed.

The corrosion rates of ZnAlMg coatings determined at Russia-based CS can be compared with the those of coatings of similar composition obtained in two-year tests at various locations around the world with atmospheric corrosivity categories C2 and C3. Data on the corrosion resistance of ZnAlMg coatings in various regions of the world, which are systematized and summarized in [19], are partially presented in Table 9. This table displays the corrosion rate ranges (σ, µm/year) of ZnAlMg coatings after 2 years of exposure to atmospheres with corrosion categories C2 and C3 for zinc. It can be seen that σ varies in a wide range for ZnAlMg coatings with different compositions.

The corrosivity of the atmosphere at the CS in Russia also corresponded to categories C2 and C3. For comparison with data displayed in Table 8, the corrosion rate *σ*_2_, g m^−2^ year^−1^ was converted to σ_2_, μm year^−1^ using the specific density of the coatings, Table 10.

The intervals of σ_2_ values at the CS in Russia with atmospheric category C2 (Table 10) indicate that the corrosion rates of the ZnAlMg coatings under study are mainly comparable to those observed in various regions of the world. The values of *σ*_2_ at NCS (Category C3) also match the ranges in the corrosion rate presented in Table 9. It can be expected that with a longer exposure of ZnAlMg coatings, their corrosion rate will decrease to a greater extent than that of HDG coatings.

### 3.5. Approximate Service Life of Coatings

For the structures and facilities whose load-bearing capacity should be maintained for a long time while the emergence of rust on the surface is of minor importance, the service life (*τ*_sl_, year) in the first approximation can be estimated using the time of full coating dissolution. The corrosion losses of zinc and other standard metals usually decrease for 6–8 years prior to the onset of a stationary stage [33]; therefore, the time of full coating dissolution can be accurately estimated only in long-term tests of samples.

The approximate service life (*τ*_sl_) estimates until complete dissolution of HDG and ZnAlMg coatings were calculated using the two-year corrosion rate. The calculations were carried out using Equation (5):*T* = M/σ(5)
where M is the mass of the coating, g m^−2^, and σ is the corrosion rate, g m^−2^ year^−1^. The nominal values of M for the coatings are presented in Table 1.

The values of *T* for each CS are displayed in Table 11. The atmosphere corrosivity categories according to the categories developed for zinc were determined using the *K*_1_ values for the coatings.

It is possible that the values of *T* given in Table 11 are underestimated, since the corrosion rates of coatings decrease with time. More accurate values of the service lives of HDG and ZnAlMg coatings can be obtained from calculations using the corrosion rates of coatings obtained in tests at least 4 years long.

## 4. Conclusions

The values of four one-year corrosion losses (*K*_1_) of steel 08ps, zinc, ZnAlMg, and HDG coatings with the setting of samples at the beginning of each season in two locations with marine atmospheres and two locations with non-marine atmospheres in an open area and under a shelter differ significantly, though the average annual atmospheric parameters are almost the same. To clarify the reasons for this effect, additional studies are needed that would take the meteorological and aerochemical parameters of the first months of testing into account.It has been shown that the average corrosion losses of materials for the four batches of samples for the first year in an open area is greater than under a shelter, except for steel 08ps at NCS and FECS, as well as ZnAlMg coatings at NCS. After two years in an open area, the corrosion losses at the CS with a marine atmosphere are smaller than or comparable to the corrosion losses under a shelter. This may be due to the accumulation of sea salts on the surface of materials in the absence of the wash-off effect of rains.The first-year corrosion losses of materials were used to determine the atmosphere corrosivity categories in relation to each material in open area and under a shelter. For ZnAlMg and HDG coatings, the atmosphere corrosivity categories developed for zinc were used. It has been shown that the corrosiveness of the atmosphere defined based on four values of K_1_ may correspond to two different categories or different subcategories of category C2.To determine the K_1_ values without conducting repeated one-year tests, dose–response functions for zinc, ZnAlMg, and HDG coatings for exposure conditions in an open area and under a shelter have been suggested. The errors in the *K*_1_ values calculated for each material do not exceed the allowable uncertainty ranges set by ISO 9223:2012.Based on the corrosion rates of ZnAlMg and HDG coatings over 2 years, an approximate estimate of their service life has been found.

## Figures and Tables

**Figure 1 materials-16-05214-f001:**
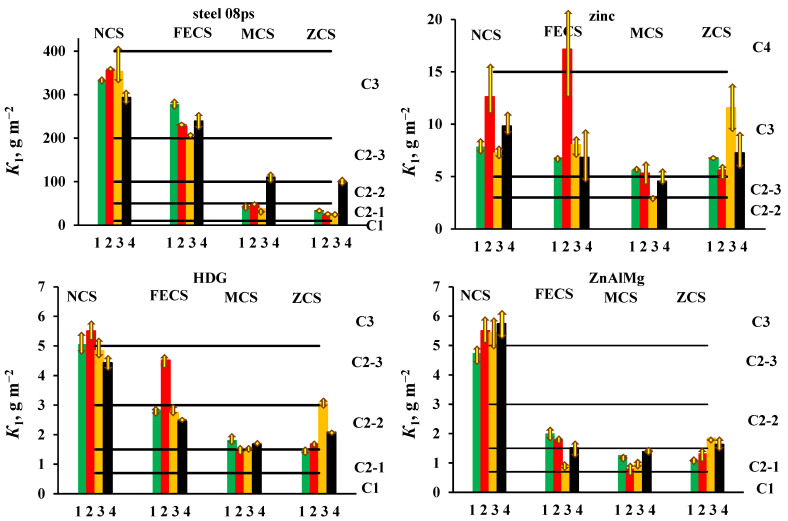
Open area. Atmosphere corrosivity categories for 4 one-year periods of exposure at each CS: spring (1▐), summer (2▐); fall (3▐); winter (4▐).

**Figure 2 materials-16-05214-f002:**
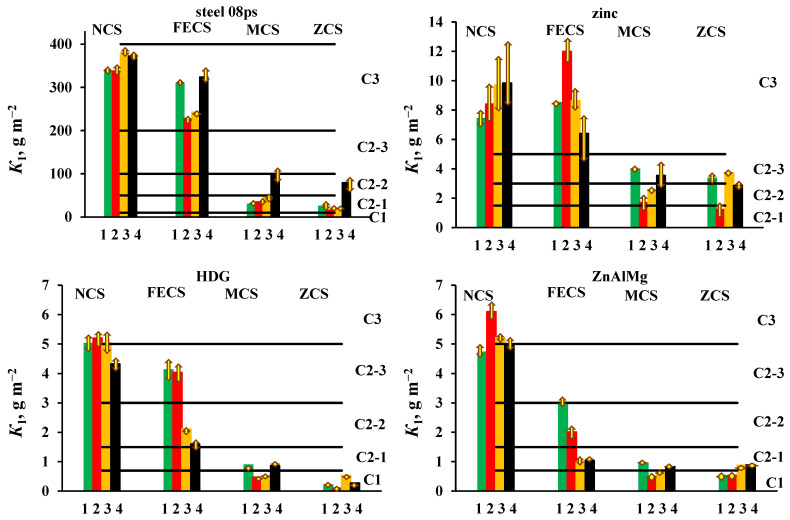
Shelter. Atmosphere corrosivity categories for 4 one-year periods of exposure at each CS: spring (1▐), summer (2▐); fall (3▐); winter (4▐).

**Figure 3 materials-16-05214-f003:**
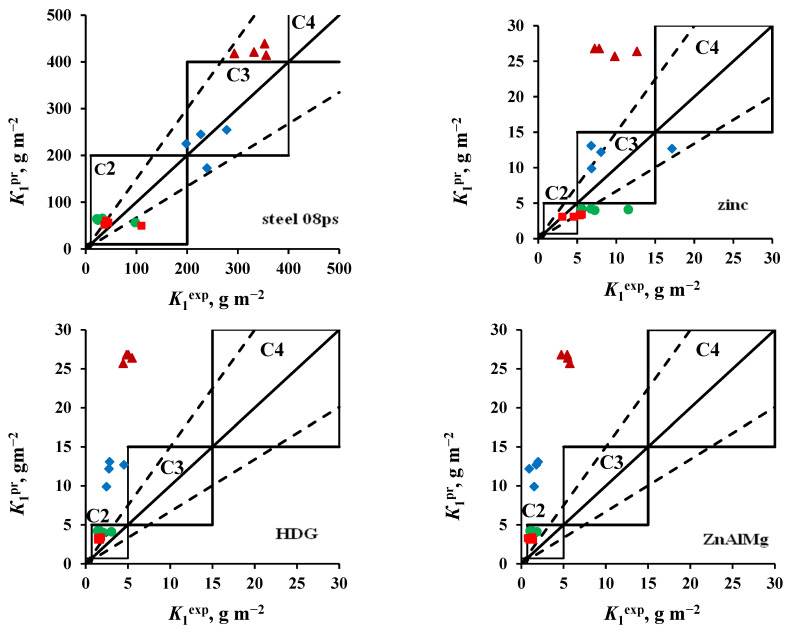
Open area. Comparison of the corrosivity categories determined using *K*_1_^ex^ and estimated from *K*_1_^pr^ calculated using Equation (1) for steel 08ps and using Equation (2) for the other materials, for ZCS (●), MCS (■), FECS (♦), and NCS (▲).

**Figure 4 materials-16-05214-f004:**
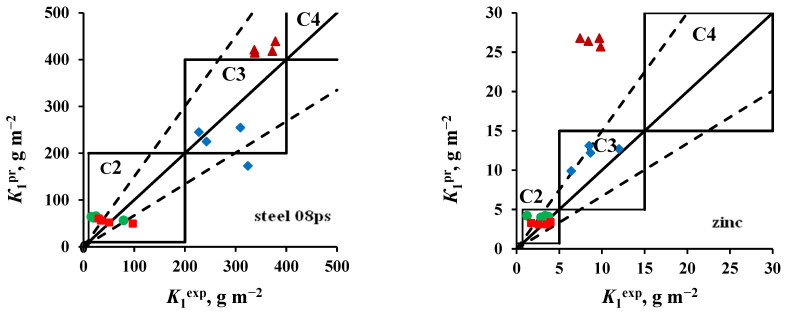
Shelter. Comparison of the corrosivity categories determined using *K*_1_^ex^ and estimated from *K*_1_^pr^ calculated using Equation (1) for steel 08ps and using Equation (2) for the other materials, for ZCS (●), MCS (■), FECS (♦), and NCS (▲).

**Figure 5 materials-16-05214-f005:**
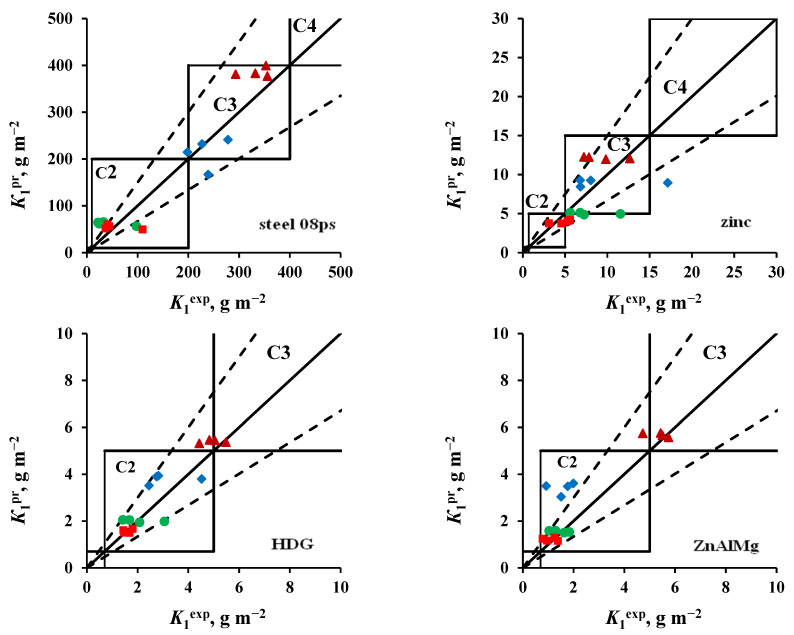
Open area. Comparison of the corrosivity categories determined from *K*_1_^ex^ and estimated using *K*_1_^pr^ values calculated using the new *A* and *β* coefficients (Table 6) for ZCS (●), MCS (■), FECS (♦), and NCS (▲).

**Figure 6 materials-16-05214-f006:**
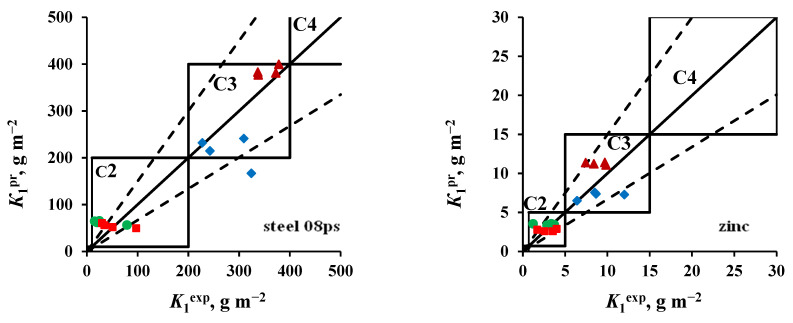
Shelter. Comparison of the corrosivity categories determined from *K*_1_^ex^ and estimated using *K*_1_^pr^ values calculated using the new *A* and *β* coefficients (Table 6) for ZCS (●), MCS (■), FECS (♦) и NCS (▲).

**Table 1 materials-16-05214-t001:** Elemental composition and nominal mass of coatings.

Coating	Approx. Composition (wt %)		Nominal Coating Mass, g/m²
	Al	Mg	
ZnAlMg	1.0–1.4	1.0–1.4	275
HDG	0.18–0.30	-	120

**Table 2 materials-16-05214-t002:** Location of corrosion stations and atmosphere types.

Corrosion Station	Designation	Longitude/Latitude	Classification
Moscow	MCS	55°65′ N/37°54′ E	urban
Zvenigorod (Moscow region)	ZCS	55°04′ N/37°57′ E	rural
Dal’nie Zelentsy (Barents Sea shore)	NCS	69°07′ N/36°04′ E	marine
Vladivostok (Sea of Japan shore)	FECS	43°04′ N/131°57′ E	marine

**Table 3 materials-16-05214-t003:** Annual average meteorological and chemical parameters of atmosphere at CS for year-long test periods.

Parameter	Batch No.				Average Value
	1	2	3	4	
NCS					
*T*, °C	2.1	2.1	2.7	2.7	2.4
*RH*, %	78	78	79	80	79
Prec, mm/year	389	374	317	220	325
[SO_2_], mg/(m^2^·day)	2.6	2.53	2.55	2.50	2.55
[Cl^−^], mg/(m^2^·day)	187	184	183	170	181
FECS					
*T*, °C	7.1	7.1	6.8	6.5	6.9
*RH*, %	66	66	66	67	66
Prec, mm/year	666	625	884	771	737
[SO_2_], mg/(m^2^·day)	3.63	3.27	3.28	3.33	3.38
[Cl^−^], mg/(m^2^·day)	36.3	34.8	24.8	14.4	27.6
MCS					
*T*, °C	8.2	7.6	7.5	7.0	7.6
*RH*, %	67	67	66	68	67
Prec, mm/year	665	559	438	421	521
[SO_2_], mg/(m^2^·day)	2.85	3.0	2.96	2.70	2.89
ZCS					
*T*, °C	6.4	6.0	5.9	4.9	5.8
*RH*, %	82	81	80	81	81
Prec, mm/year	533	479	459	456	482
[SO_2_], mg/(m^2^·day)	2.46	2.88	2.91	2.71	2.74

**Table 4 materials-16-05214-t004:** Corrosion losses of materials (*K*_1_, g m^−2^) for one-year periods of exposure, difference in corrosion losses of samples from the average value of *K*_1_ (±∆*K*_1_, g m^−2^) for each batch, and category of corrosive aggressiveness of the atmosphere in an open area.

Material	Batch	*K*_1_, g m^−2^												Categories			
	No.	NCS			FECS			MCS			ZCS			NCS	FECS	MCS	ZCS
		*K* _1_	−∆*K*_1_	+∆*K*_1_	*K* _1_	−∆*K*_1_	+∆*K*_1_	*K* _1_	−∆*K*_1_	+∆*K*_1_	*K* _1_	−∆*K*_1_	+∆*K*_1_				
Steel 08ps	1	331.9	−4.3	+6.7	277.8	−8.7	+9.7	40.4	−4.7	+2.8	32.8	−0.9	+1.4	C3	C3	C2-1	C2-1
	2	355.8	−1.4	+1.8	226.8	−1.4	+0.8	44.5	−0.5	+0.4	22.1	−0.9	+1.6	C3	C3	C2-1	C2-1
	3	352.8	−28.2	+52.1	198.1	−2.0	+2.9	36.9	−0.5	+0.7	24.6	−0.3	+0.2	C3	C2-3	C2-1	C2-1
	4	293.1	−13.2	+10.5	239	−26.4	+16.7	110	−6.7	+6.3	97.8	−16.9	+9.1	C3	C3	C2-3	C2-2
	Average	333.4			235.4			57.9			44.3			C3	C3	C2-2	C2-1
Zn	1	7.81	−0.62	+0.82	6.78	−0.40	+0.21	5.58	−0.31	+0.22	6.79	−0.11	+0.06	C3	C3	C3	C3
	2	12.6	−1.52	+2.89	17.14	−3.79	+4.59	5.31	−1.07	+1.20	5.6	−0.50	+0.95	C3	C4	C3	C3
	3	7.23	−0.54	+0.77	8.03	−1.13	+0.96	3.09	−0.26	+0.16	11.55	−2.78	+1.68	C3	C3	C2-3	C3
	4	9.81	−1.09	+1.24	6.82	−2.05	+2.55	4.54	−0.57	+1.07	7.25	−1.32	+1.96	C3	C3	C2-3	C3
	Average	9.36			9.69			4.63			7.80			C3	C3	C2-3	C3
HDG	1	5.04	−0.27	+0.30	2.82	−0.13	+0.11	1.8	−0.14	+0.17	1.42	−0.16	+0.12	C3	C2-2	C2-2	C2-1
	2	5.5	−0.28	+0.24	4.52	−0.19	+0.22	1.44	−0.13	+0.12	1.68	−0.10	+0.05	C3	C2-3	C2-1	C2-2
	3	4.83	−0.22	+0.33	2.75	−0.09	+0.21	1.47	−0.10	+0.10	3.06	−0.13	+0.10	C2-3	C2-2	C2-1	C2-3
	4	4.43	−0.27	+0.13	2.45	−0.05	+0.04	1.68	−0.07	+0.08	2.08	−0.03	+0.03	C2-3	C2-2	C2-1	C2-2
	Average	4.95			3.14			1.60			2.06			C2-3	C2-3	C2-2	C2-2
ZnAlMg	1	4.73	−0.30	+0.18	1.99	−0.22	+0.16	1.25	−0.15	+0.09	1.04	−0.13	+0.07	C2-3	C2-2	C2-1	C2-1
	2	5.5	−0.34	+0.36	1.77	−0.19	+0.10	0.78	−0.16	+0.16	1.3	−0.19	+0.24	C3	C2-2	C2-1	C2-1
	3	5.43	−0.31	+0.56	0.92	−0.11	+0.16	0.96	−0.10	+0.14	1.84	−0.08	+0.07	C3	C2-1	C2-1	C2-2
	4	5.74	−0.38	+0.35	1.51	−0.21	+0.22	1.38	−0.11	+0.13	1.63	−0.18	+0.18	C3	C2-2	C2-1	C2-2
	Average	5.35			1.55			1.09			1.45			C3	C2-2	C2-1	C2-1

**Table 5 materials-16-05214-t005:** Corrosion losses of materials (*K*_1_, g m^−2^) for one-year periods of exposure, difference in corrosion losses of samples from the average value of *K*_1_ (± ∆*K*_1_, g m^−2^) for each batch, and category of corrosive aggressiveness of the atmosphere under a shelter.

Material	Batch	*K*_1_, g m^−2^												Categories			
	No.	NCS			FECS			MCS			ZCS			NCS	FECS	MCS	ZCS
		*K* _1_	−∆*K*_1_	+∆*K*_1_	*K* _1_	−∆*K*_1_	+∆*K*_1_	*K* _1_	−∆*K*_1_	+∆*K*_1_	*K* _1_	−∆*K*_1_	+∆*K*_1_				
Steel 08ps	1	336.9	−4.2	+3.5	309.3	−1.3	+1.1	29.6	−0.59	+0.81	24.8	−6.23	+6.12	C3	C3	C2-1	C2-1
	2	337.7	−8.9	+8.6	227.5	−5.9	+5.6	34.6	−0.64	+1.16	15.5	−0.82	+0.53	C3	C3	C2-1	C2-1
	3	378.3	−7.8	+4.7	242.5	−2.9	+1.5	50.6	−1.41	+1.52	20.0	−0.10	+0.06	C3	C3	C2-2	C2-1
	4	372.2	−5.7	+7.5	323.9	−11.1	+16.9	97.0	−17.7	+10.6	79.2	−18.5	+13.9	C3	C3	C2-2	C2-2
	Average	356.3			275.8			53.0			34.9			C3	C3	C2-2	C2-1
Zn	1	7.43	−0.50	+0.59	8.51	−0.07	+0.08	3.98	−0.11	+0.14	3.36	−0.46	+0.31	C3	C3	C2-3	C2-3
	2	8.40	−0.88	+1.67	11.99	−0.61	+0.96	1.69	−0.57	+0.39	1.22	−0.37	+0.64	C3	C3	C2-2	C2-1
	3	9.69	−1.24	+2.46	8.64	−0.62	+0.76	2.57	−0.17	+0.30	3.73	−0.17	+0.26	C3	C3	C2-2	C2-3
	4	9.84	−1.85	+2.47	6.40	−1.81	+1.35	3.54	−1.08	+0.71	2.86	−0.25	+0.32	C3	C3	C2-3	C2-2
	Average	8.84			8.89			2.95			2.79			C3	C3	C2-2	C2-2
HDG	1	5.02	−0.26	+0.26	4.13	−0.21	+0.32	0.89	−0.05	+0.03	0.21	−0.01	+0.02	C3	C2-3	C2-1	C1
	2	5.20	−0.26	+0.19	4.04	−0.26	+0.24	0.47	−0.05	+0.05	0.1	−0.04	+0.04	C3	C2-3	C1	C1
	3	5.03	−0.34	+0.38	2.11	−0.12	+0.09	0.52	−0.05	+0.07	0.53	−0.07	+0.11	C3	C2-2	C1	C1
	4	4.33	−0.23	+0.12	1.6	−0.18	+0.13	0.86	−0.08	0.05	0.27	−0.04	+0.03	C2-3	C2-2	C2-1	C1
	Average	4.90			2.97			0.69			0.28			C2-3	C2-2	C1	C1
ZnAlMg	1	4.73	−0.13	+0.24	3.0	−0.12	+0.20	0.96	−0.04	+0.06	0.53	−0.08	+0.09	C2-3	C2-2	C2-1	C1
	2	6.10	−0.31	+0.26	2.01	−0.21	+0.16	0.5	−0.12	+0.14	0.53	−0.04	+0.07	C3	C2-2	C1	C1
	3	5.22	−0.16	+0.13	1.0	−0.12	+0.04	0.65	−0.06	+0.07	0.88	−0.05	+0.03	C3	C2-1	C1	C2-1
	4	4.94	−0.24	+0.16	1.04	−0.11	+0.09	0.83	−0.04	+0.04	0.89	−0.03	+0.01	C2-3	C2-1	C2-1	C2-1
	Average	5.25			1.76			0.74			0.71			C3	C2-2	C2-1	C2-1

**Table 6 materials-16-05214-t006:** Values of DRF coefficients for materials exposed in open area and under a shelter.

Coefficient	Zn	HDG	ZnAlMg
	Open Area		
*A*	0.55	0.22	0.17
*β*	0.27	0.30	0.38
	Shelter		
*A*	0.38	0.12	0.12
*β*	0.35	0.46	0.46

**Table 7 materials-16-05214-t007:** Relative errors in calculated *K*_1_^pr^ values.

Error	Open Area				Shelter			
	Steel 08ps	Zn	HDG	ZnAlMg	Steel 08ps	Zn	HDG	ZnAlMg
	In comparison with *K*_1_^ex^ values obtained for each of 4 batches							
−δ%	35.1	29.3	12.9	10.9	31.9	18.2	18.8	4.8
+δ%	29.2	40.4	24.7	42.6	41.0	21.0	33.1	41.6
	In comparison with the average *K*_1_^ex^ values obtained for 4 batches							
−δ%	7.0	17.5	2.4	-	21.3	8.1	2.7	-
+δ%	27.6	44	15.5	8.1	29.4	12.5	19.0	25.6

**Table 8 materials-16-05214-t008:** Corrosion rate *σ*_2_, g m^−2^ year^−1^ for 2 years of exposure in open area and under a shelter.

Material	NCS		FECS		MCS		ZCS	
	OA	Shelter	OA	Shelter	OA	Shelter	OA	Shelter
Steel 08ps	194.1	224.4	177.7	245.6	35.5	28.2	24.7	21.6
ZnO	4.95	5.81	4.99	4.70	4.14	2.75	4.26	1.93
HDG	3.83	3.76	2.42	2.54	1.43	0.56	1.31	0.26
ZnAlMg120	3.52	3.40	1.77	2.42	0.90	0.60	0.75	0.44

**Table 9 materials-16-05214-t009:** Corrosion rates (*σ*_2_) of ZnAlMg coatings after two years of exposure in open atmospheres of categories C2 and C3 at various locations around the world.

References	Coating	σ_2_, μm year^−1^	
		C2	C3
[12]	Zn1%Al1%Mg Zn1.5%Al1.5%Mg	0.22–0.26 Average 0.24	0.39–0.81 Average 0.6
[12]	Zn2%Al2%Mg	0.15	0.4–0.64 Average 0.52
[16]	Zn2.67%Al1.51%Mg Zn2.71%Al1.51%Mg Zn2.72%Al1.49%Mg Zn2.9%Al1.6%Mg	-	0.34–0.54 Average 0.41
[17,18]	Zn3.7%Al3%Mg	0.09–0.36 Average 0.23	0.42–0.81 Average 0.62

**Table 10 materials-16-05214-t010:** Corrosion rates (*σ*_2_) of HDG and ZnAlMg coatings after 2 years of testing at the CS located in Russia.

Testing Site	Category	Testing Conditions	σ_2_, μm year^−1^	
			HDG	ZnAlMg
ZCS, MCS FECS	C2	OA	0.19–0.35 Average 0.27	0.12–0.27 Average 0.20
		Shelter	0.04–0.36 Average 0.20	0.07–0.38 Average 0.23
NCS	C3	OA	0.55	0.54
		Shelter	0.54	0.53

**Table 11 materials-16-05214-t011:** Approximate estimates of service life (*T*, year) of HDG and ZnAlMg coatings calculated from the corrosion losses for 2 years of exposure.

Coating	ZCS		MCS		FECS		NCS	
	Category	*T*	Category	*T*	Category	*T*	Category	*T*
	Open area							
HDG	C2	˃200	C2	190	C2	113	C3	72
ZnAlMg	C2	160	C2	130	C2	68	C3	34
	Shelter							
HDG	C2	˃˃200	C2	˃˃200	C2	108	C3	73
ZnAlMg	C2	˃200	C2	200	C2	50	C3	35

## Data Availability

Not applicable.

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
