# Peer review of "Corrosion Resistance of Zinc and Zinc-Aluminum-Magnesium Coatings in Atmosphere on the Territory of Russia"

_materials, 2023, doi:10.3390/ma16155214_

Round 1

Reviewer 1 Report

Please find my comments enclosed in the attached file. 

I have two major comments that must be addressed:

1) The number of parallels have not been given. In Table 4 and 5, and Figure 1 and 2, the standard deviation for the results must be given. If there were no parallels, the four batches must be regarded as parallels, and the effect of season when the samples were deployed cannot be discussed. 

2) The conclusions must be revised. See my comments in the file. All conclusions must have been discussed in the paper. Conclusion 1 appears as speculation, unless the statistical evidence is presented. Conclusion 4 is not substantiated by the discussion. Conclusion 5 is too obvious - this is the purpose with corrosivity categories.

To be considered: 

1) The title of the paper (and the paper in general) emphasise the country where the study was performed. I think this is less important. It's the exposure conditions that determines the corrosivity category, independent of country. A better title would be "ATMOSPHERIC CORROSION RESISTANCE OF ZINC AND ZINC–2 ALUMINUM–MAGNESIUM COATINGS"

2) Section 3.4 should be split into one part discussing the 2 years results, and one part discussing coating lifetime. 

Reviewer 2 Report

Very useful piece of work, The authors collected a lot of data which are helpful for both coatings raw material producters and coating formulators. Thanks.

Author Response

The authors are grateful to the Reviewer for the positive evaluation of their work.

Reviewer 3 Report

Remarks: In this manuscript, the direction of the research was to assess the corrosion resistance of ZnAlMg coatings in four representative locations in Russia, also, for determine the categories of atmosphere corrosivity, develop the dose-response function (DRF) and to roughly estimate the service life of the coatings.

The article is very complex and interest of Materials journal, but specific aspects mentioned in the following require the minor revision of the paper:

In Abstract the sentence between lines 14-15 should be rephrased, because Zn-Al-Mg alloys have been on the market for more than 15 years;

1. Introduction:

The bibliographies in the first part of the introduction (sources1-9) are approximately 15 years old. In order to mention corrosion protection mechanisms or data on corrosion of Zn alloy coatings you should mention current literature sources. The idea of an introduction is to show the current status and to elaborate on the novelty of your work and have engagement with the sources.

2.1. Materials

You should describe thoroughly the deposition of the Zn-Al-Mg coating (e.g. thickness, density, etc), possibly the microstructures in cross section. The procedure applied should be briefly exemplified because not many readers have access to the standard you indicated. You should also present the technique by which you removed the corrosion products.

3.3. Accessing the atmosphere corrosivity categorie

The self-citations from line 224 should be inserted in the part 2. materials and methods considering that your research is more extensive and based on the results from that sources.
